The regulating effects and mechanism of biochar and maifanite on copper and cadmium in a polluted soil-Lolium perenne L. system

Ding Yuan 39011@nchu.edu.cn
Wang Weiya
Ao Shiying
National-Local Joint Engineering Research Center of Heavy Metal Pollutant Control and Resource Utilization, School of Environmental and Chemical Engineering, Nanchang Hangkong University , Nanchang , China
Kah Melanie
Electronic publication date: 2021 Aug 9
Publication date: 2021
Volume: 9
Electronic Location ID: e11921
Received 2021 Mar 5; Accepted 2021 Jul 16
Copyright: ©2021 Ding et al.
Copyright year: 2021
Copyright holder: Ding et al.
License: This is an open access article distributed under the terms of the Creative Commons Attribution License, which permits unrestricted use, distribution, reproduction and adaptation in any medium and for any purpose provided that it is properly attributed. For attribution, the original author(s), title, publication source (PeerJ) and either DOI or URL of the article must be cited.
License URL: https://creativecommons.org/licenses/by/4.0/

Keywords: Dregs Biochar, Maifanite, Immobilization, Soil colloid, Cu, Cd

Funding: National Natural Science Foundation of China No. 41967021 Key Research and Development Program of Jiangxi province No. 20181ACG70021 The Program for Education Department of Jiangxi Province No. GJJ180528 This work was supported by the Program for National Natural Science Foundation of China (No. 41967021), the Key Research and Development Program of Jiangxi province (No. 20181ACG70021), and the Program for Education Department of Jiangxi Province (No. GJJ180528). The funders had no role in study design, data collection and analysis, decision to publish, or preparation of the manuscript.

==============================
Arable land polluted by copper (Cu) and cadmium (Cd) is a widespread problem. The use of biochar and/or clay mineral as a soil amendment can effectively solidify heavy metals in the soil. We applied biochar (BC), iron modified biochar (Fe-BC), maifanite (MF, a kind of clay minerals), a combination of BC with MF (BC:MF), and Fe-BC with MF (Fe-BC:MF) at a 2 wt % dose as soil amendments to study their ability to prevent Cu and Cd from accumulating in ryegrass (Lolium perenne L.). We found that after 90 days of cultivation, the Cd and Cu content both significantly decreased in ryegrass shoots from 2.06 and 209.3 mg kg−1 (control) to 1.44–2.01 and 51.50–70.92 mg kg−1, respectively, across treatments (p < 0.05). Similarly, the bioconcentration factor (BCF) for Cd/Cu was significantly smaller (P < 0.05) in all amendments versus control soil. This trend differed among the shoot, BCF, and transportation factor (TF). Combining BC:MF or Fe-BC:MF did not significantly improve the Cd/Cu stabilization in the soil compared to the corresponding single amendment (p > 0.05). Our adsorption balance experiment showed that BC, Fe-BC, and MF physically and chemically adsorbed Cd and Cu by complexation with functional groups (mesoporous nanomaterials) whose porosity measurements ranged from 0.68 to 78.57 m2 g−1. Furthermore, the amorphous crystalline iron oxide binding Cd and Cu was the key to immobilizing these metals in the soil. The amendments applied in our study show promise for enhancing immobilization of Cu and Cd in contaminated paddy soils.

Introduction

Soil pollution from heavy metals and metalloids is a worldwide concern that affects food production and human health. Anthropogenic activities, such as mining, smelting, military operations, electronic industries, fossil fuel consumption, waste disposal, agrochemical use, and irrigation are the key factors contributing to the increase of heavy metal contents in the soil (Barsova et al., 2019; Liu et al., 2018). Survey data has shown that approximately 19.4% of the total arable land in China has been polluted by various contaminants, among which Cd is highest (Li et al., 2019). The soil in southern China has been seriously contaminated with Cu and Cd near Cu smelters, resulting in large stretches of barren land. It is critical to remediate the polluted soil, restore the natural ecosystem, and protect human health.

In-situ chemical immobilization, also known as in-situ solidification/stabilization, is considered to be an inexpensive remediation method to immobilize heavy metals. This method uses chemical reagents, natural minerals, and environmentally friendly waste to decrease heavy metal mobility and phytoavailability. Although the method does not reduce the total amount of heavy metals in the soil, it does minimize the potential transportation of heavy metals from soil to plants (Song et al., 2017). The use of chemical immobilization is still an effective strategy to remediate large areas of contaminated farmland (Shen et al., 2019).

Inorganic and organic materials were used extensively in the in situ chemical immobilization of heavy metals. Clay mineral materials, such as sepiolite, smectite, zeolite, and bentonite, are the most effective inorganic amendments to adsorb heavy metals in soil. These can significantly reduce the phytoavailability of heavy metals due to their unique coordination and Si-O tetrahedron and Al-O octahedron structures’ adsorption ability (Khalid et al., 2017; Sun et al., 2016). Clay minerals are advantageous in the remediation of heavy metals in soil for their low dosage and high effectiveness. However, there are risks when changing the soil’s physical and chemical properties. Recent research has shown that organic amendments prepared from low-cost waste materials decreased the mobility or phytoavailability of heavy metals and improved the soil fertility (Li et al., 2019; Quan et al., 2020). For example, biochar produced from waste biomass is a typical amendment due to their higher cation exchange capacities (CEC), chemical functional groups, microporous structure, and high pH (Zhai et al., 2018). In acidic soils, the ability to immobilize Cd and Cu was related to soil pH and the number of oxygen-containing groups on the surface of the biochar. Biochar materials were further modified in a variety of ways, such as loading iron hydroxyl group to enhance the surface adsorption capacity (Wan, Li & Parikh, 2020; Zhou et al., 2018).

It appears that combining clay minerals with organic amendments, such as biochar, is more suitable for treating contaminated soils. However, the immobile capacity of heavy metals varies greatly with different combinations and applications of organic and inorganic amendments. For instance, Zhang & Ding (2019) investigated the use of hydroxyapatite, bentonite, and biochar, alone or in combination, for remediating Cd-Pb-contaminated soil. Their results showed that the largest reduction in the soluble fraction of Cd and Pb acids was observed under combined amendments. Cu and Cd’s phytoavailability and transfer mechanisms in polluted soil still need to be studied with different types, and combinations of, clay minerals and biochar.

The accumulation of heavy metals in plants, especially in the shoots, was a direct indicator of the phytoavailability of heavy metals in the soil. However, there were significant differences in amounts due to variations in plant types, as well as the effect of operability. The extraction concentration by a chemical reagent was used to indicate the phytoavailability of heavy metals. It is generally believed that the phytoavailability of Cu and Cd decreases when the weak acid soluble fraction oxidizable and transferred to residual fractions. Soil colloids are an important carrier that can render heavy metals immobile in polluted soils (Tang, Katou & Suzuki, 2020). Cu and Cd content and fractions in colloids are the key to explaining the immobilization effect and mechanism. Little attention has been paid to the influence of soil colloids on the redistribution and phytoavailability of Cu and Cd in polluted soil. We sought to (1) compare the remediation effects of different types and combinations of biochar and maifanite (a natural clay mineral); (2) understand these amendments’ adsorption ability; and (3) elucidate how soil colloids remediate Cu and Cd in polluted soil.

Materials & Methods

Soil

The soil samples were collected from the plough layer (0–20 cm deep) of a paddy field near a smelter in Jiangxi Province, China (117°12′34″E and 28°19′6″N). The soil was air dried at room temperature and passed through a 10 mm sieve to remove stones and roots. The soil’s basic physical and chemical properties were: pH: 4.96, soil organic matter (SOM): 2.0%, cation exchange capacity (CEC): 68.3 mmol kg−1, Cu concentration: 240.0 mg kg−1, Cd concentration: 1.93 mg kg−1, Fe concentration: 17.20 mg kg−1, available phosphorous (AP): 5.82 mg kg−1, and available nitrogen (AN): 141.4 mg kg−1.

Biochar testing

Biochar was produced from Radix isatidis (a component in traditional Chinese medicine) residue. R. isatidis was purchased from a pharmacy. The specific preparation method was as follows: deionized (DI) water at a ratio of 500 mL, to which 50 g radix Isatidis was added and boiled for approximately 30 min. The dregs were removed and placed in an oven at 105 °C for 24 h. They were then dried and ground into powder in an agate bowl for later use. A laboratory-scale gas flow-controlled tube-type furnace was used to pyrolyze the R. i satidis dreg powder at 500 °C for 4 h under N2 at a flow rate of 1,000 mL min−1to produce biochar. Biochar was further pulverized to 0.25 mm (60 mesh) and marked as BC. The basic properties of final biochar are presented in Table 1.

Table 1 Basic properties of biochar and maifanite.

Treatment	pH	Cu/mg kg−1	Cd/mg kg−1	Fe/g kg−1	Ash content/%	SSA/m2 g−1	
BC	10.93 ± 0.09	18.49 ± 0.59	ND	4.66 ± 0.51	13.73 ± 0.62	9.45	
Fe-BC	8.44 ± 0.03	26.80 ± 0.94	ND	172.9 ± 5.0	26.24 ± 1.26	78.57	
MF	7.03 ± 0.18	12.89 ± 0.53	ND	ND	ND	0.68	
Notes.

BC biochar from medicine residue

Fe-BC iron modified biochar

MF maifanite

“ND” indicates undetected or below the detection limit. Values were described as means ± standard deviation (n = 3).

Modified biochar and maifanite

We soaked 20 grams of BC in 1 mol L−1 FeCl3 solution, stirred for 10 min, let it sit for 24 h, then filtered and dried it in an 80 °C oven. The dried BC was put into muffle furnace for 1 h at 500 °C. After washing, drying, and passing through 60 mesh sieves (0.25 mm), the iron-modified BC was produced and named Fe-BC. The basic properties of Fe-BC are shown in Table 1.

Maifanite, a natural SiO2 and NaAlSi3O8 rich mineral, was obtained from a water purification factory using a mesh size of 60–80 (0.18–0.25 mm) and named MF (Table 1).

Pot experiment

Six treatments were carried out in triplicate for the pot experiment as below: (1) CK, polluted soil as control, (2) BC, Fe-BC, MF, biochar and maifanite BC: MF (1:1, w/w), iron modified biochar and maifanite Fe-BC: MF (1:1, w/w).

A sample of CK (2.00 kg) was completely mixed with amendments (20 g kg−1 soil) and transferred to a plastic pot (height: 15 cm; diameter: 17 cm) with deionized (DI) water in a greenhouse at 25 °C. Water content was maintained at 65% of the water holding capacity. No fertilizer was applied to any treatments during the incubation period. Forty ryegrass (Lolium perenne L.) seeds were spread in each pot 40 days after cultivation. After germination, 20 seedlings remained in each pot. The shoots were cut off after 90 days. Soil colloid was extracted from the soil to analyze the fraction distribution of heavy metals after the pot experiment.

Morphology and fraction distribution of amendments before and after adsorbing Cu and Cd

BC, Fe-BC and MF amendments measuring 0.1 g each were passed through a 100-mesh sieve placed in a 50 mL stoppered conical flask. Then, 20 mL standardized solutions of 50 mg L−1 Cu and Cd were added to the flask. We added a 0.01 mol L−1 NaCl solution to simulate soil conditions, followed by the addition of 0.01 mol L−1 HCl and 0.01 mol L−1 NaOH solutions to adjust the pH value to 4.96, which was the pH value of the soil. Samples were kept at a constant room temperature for 24 h before centrifugation and filtration. The amendments were dried in an oven at 60 °C after the adsorption of Cu and Cd and crushed in an agate bowl to a particle size of 100 mesh for heavy metal morphology and fraction distribution analysis.

Cu and Cd content in ryegrass (L. perenne L.)

The air-dried ryegrass samples were ground into powder using a pulverizer to detect the heavy metal content. Next Cu and Cd content were measured using microwave digestion combined with flame-graphite furnace atomic absorption spectrometry. Cu and Cd were then measured using the FAAS/GFAAS method (GB/T 17138-1997 and GB/T 17141-1997).

Fraction analysis of heavy metals in soil and soil colloid

Soil heavy metal fraction analysis can be divided into a weak acid soluble fraction, a reducible fraction, an oxidizable fraction, and a residue fraction according to the BCR sequential extraction method. The modified BCR sequential extraction procedures were used for heavy metal fraction analysis in the soil (Zhang & Ding, 2019). Heavy metal fractions in soil colloid can be classified as an amorphous iron oxide bound fraction, a crystalline iron oxide bound fraction, and a residue fraction (Yin et al., 2016).

Amendment microcharacters before and after adsorption

Modified BCR sequential extraction procedures were used for heavy metal fraction analysis in the amendments (Zhang et al., 2019).

Amendments’ specific surface area (SSA) values were measured using a Tristar II type 3020 (V1.04) analyzer (Micromeritics Instrument, USA). A scanning electron microscope (SEM, SU1510, Hitachi, Japan) was used to observe the surface morphology and microstructure. The samples were mounted on a carbon stub and were coated with Au before the examination. The crystal structures of the amendments were examined using the X-ray diffraction analysis technique (XRD, D8 Advance X-ray diffractometer; Bruker) before and after adsorption. The surface functional groups of biochar and maifanite were determined using the Fourier transform infrared spectroscopy technique (FTIR, Vertex 70, Bruker, Germany).

Calculation and statistical analysis

The bioconcentration factor (BCF) and the transportation factor (TF) were calculated using the following equations (Zhang et al., 2020):

(1) BCF=Cshoot+CrootCsoil

(2) TF=CshootCroot

where Cshoot is the metal concentration in the ryegrass shoots (mg kg−1), Croot is the metal concentration in the root of ryegrass (mg kg−1), and Csoil is the metal concentration in the soil (mg kg−1).

The data were subjected to one-way analysis of variance (ANOVA) using SPSS software (version 23.0). Duncan’s multiple range test was performed to determine the difference between treatments. Means were compared using the least significant difference test at a p < 0.05 level of significance, and letters were used to suggest the significant variations in the data set. All of the results in the tables and figures were described as means ± standard deviation (n = 3).

The precision and accuracy of heavy metal analysis were assessed through repeated analysis of the samples against National Institute of Standard and Technology, Standard Reference Material (GBW07408; National Standard Detection Research Center, Beijing, China) for all the heavy metals. The results were found to be within ±2% of the certified value. Variations were found to be less than 10%.

Results and Discussion

Immobilizing different amendments on Cu and Cd in soil

Effect on ryegrass

L. perenne L. is a good indicator of heavy metal polution in soil as they are resistant to Cu and Cd. Ryegrass can enter the food chain as a kind of pasture, therefore, we analyzed Cu and Cd content change in ryegrass shoots after the addition of different amendments (Fig. 1).

Figure 1 Cu (A) and Cd (B) content in shoot of ryegrass treated with different amendments.

The data shown here was the averages of three replicates with the standard error indicated by the vertical bars. Different lowercase letters in each column represented statistically significant differences among treatments (P < 0.05).

As shown in Fig. 1A, Cu content in shoots significantly decreased from 209.3 mg kg−1 (CK) to 51.50 mg kg−1 (BC), 57.90 mg kg−1 (Fe-BC:MF), 59.29 mg kg−1 (BC:MF), 63.74 mg kg−1(MF), and 70.92 mg kg−1 (Fe-BC). Cd also decreased from 2.06 mg kg−1(CK) to 1.43 mg kg−1 (MF), 1.46 mg kg−1 (BC), 1.51 mg kg−1 (BC:MF), 1.53 mg kg−1 (Fe-BC), and 2.01 mg kg−1 (Fe-BC:MF). Cu was immobilized in the following order: BC >Fe-BC:MF ≈ BC:MF >MF ≈ Fe-BC. Cu content decreased by 75.4%, 72.3%, 71.7%, 69.6%, and 66.1%, respectively, compared with the control. By contrast, Cd immobilized in shoots after treatment with Fe-BC:MF did not differ significantly from the control (P > 0.05). The treatments BC, Fe-BC, MF, and BC:MF significantly reduced the Cd content in ryegrass shoots by 29.3%, 25.8%, 30.4%, and 26.9%, respectively.

The absorption of Cu and Cd in ryegrass shoot revealed the significant resistance effects of the amendments except for Fe-BC:MF. Furthermore, the resistance effect on Cu was much greater than on Cd according to the decreasing amplitude of the heavy metal content in the shoot. Our results was confirmed by Xu & Zhao (2013), who adopted biochar to adsorb Cu(II), Cd(II) and Pb(II) in soil from southern China, and found that the complexing ability of the biochars with Cu(II) and Pb(II) was much stronger than that with Cd(II) and induced more specific adsorption of Cu(II) and Pb(II) by the soils than that of Cd(II).

The combination amendment of Fe-BC and MF or BC and MF did not show better advantage than the corresponding sole amendment (p > 0.05).

Effect of different amendments on BCF of Cu and Cd

BCF reflects the capacity of the plant as a whole (including shoot and root) to uptake Cu and Cd from the soil. This indicator is significant in plants that enter the food chain because it fully expresses the total amount of heavy metals taken up from the soil.

The BCF value for Cu and Cd was significantly (p < 0.05) smaller in any amendment than that in the unamended soil (Fig. 2). However, the trend and reducing amplitude of the BCF value were not consistent with the corresponding heavy metal content in the ryegrass shoot. The BCF value for Cu decreased from 4.82 (CK) to 2.75–3.89 (amendments). The value decreased for Cd from 4.92 (CK) to 2.55–3.54 after the application of amendments. In addition, the best immobilization effect of the BCF value on Cu is MF, and Fe-BC of Cd, among the amendments.

Figure 2 Effects of different amendments on BCF of Cu (A) and Cd (B).

Different amendments’ effects on TF of Cu and Cd

The content of Cu and Cd in ryegrass shoots was inconsistent with the corresponding value of BCF. This effect may be due to the varying transfer capacity of Cu and Cd from root to shoot after being treated with different amendments. The changes in Cu and Cd with the transportation factor (TF) are shown in Fig. 3.

Figure 3 Effects of different amendments on transportation factor for Cu (A) and Cd (B).

TF is an important indicator of the transportation of heavy metals in plant tissues (Zhang et al., 2020). Figure 3A shows that the TF value of Cu (0.07–0.11) was significantly smaller than that of the control group (0.22), indicating that the transfer ability of Cu from the root to the shoot also decreased after the application of the amendment. However, the TF value of Cd (0.28–0.50) was significantly larger than that of the control group (0.28) (Fig. 3B).

As an essential element, copper is beneficial to the environment in trace quantities and be poisonous at high concentration. However, the total Cu content in the soil (239.98 mg kg−1) in our study was so high that ryegrass exerted a self-protection mechanism to prevent Cu transfer from root to shoot, even after amendments were applied to decrease its phytoavailability (Ali et al., 2017). Therefore, the value of BCF and TF of copper was still less than the control after the application of amendments.

Cadmium is a non-essential element for plant growth and shows carcinogenicity. A high content of Cd would inhibit plant growth or even cause death, but low content would promote plant growth. Low amounts of Cd in the soil will also promote Cd transport from root to shoot. Limited Cd content (0.30 mg kg−1) in China’s soil environmental quality standard (GB 15618-2018) is within the range required to promote plant growth and absorption. Research has shown that Cd content in test plant such as rice and peanuts is higher than food hygiene standards, while the soil Cd content is lower than the soil standard (Gallego et al., 2012). In the same way, the bioavailable Cd content after immobilization may be within a range that is easily transferred and absorbed by plants. This may be the reason that the TF of cadmium treated with the amendments increased in our study.

Microscopic characteristics of amendments and its interaction mechanism with Cu and Cd

Micromorphology and specific surface area of amendments

As described above, the sole amendment by BC, Fe-BC, MF and their combinations clearly immobilized Cd and Cu in polluted soil. We determined the amendments’ micromorphology and the fraction distribution of Cd and Cu in the amendments and soil colloid. This illuminates the microcharacteristic differences of these amendments (BC, Fe-BC, MF) and its immobilization of Cd and Cu.

The three amendments’ morphologies were examined with SEM. The BC amendment showed an irregular microstructure with an uneven surface and dense distribution of pore channels (Fig. 4A). Tiny particles attached to Fe-BC’s surface may be iron oxides (Fig. 4B), which was consistent with the data in Table 1. The content of iron increased from 4.66 g kg−1 to 172.9 g kg−1 after being modified by iron salt. The iron oxides on the surface of the biochar may be Fe3O4 or α-FeOOH, which provide abundant active hydroxyl groups for biochar (Zhu et al., 2019). The MF amendment (Fig. 3C) revealed a distinct layered structure and a large number of granular particles attached to its surface.

Figure 4 Scanning electron microscopy (SEM) micrographs of amendments BC (A), Fe-BC (B), and MF (C).

In addition to SEM, BET was also used to determine the specific surface area (SSA) of the materials. The results showed that all of the studied amendments belonged to mesoporous nanomaterials with SSA values at the range of 0.68–78.57 m2 g−1 (Table 1). The highest SSA value was detected in Fe-BC, indicating that Fe-BC could provide more adsorption of Cu and Cd than others.

Functional groups of amendments before and after adsorption of Cu and Cd

We analyzed the variations of the functional groups of the amendments before and after Cu and Cd adsorption using the FTIR spectroscopy technique (Fig. 5). The wide absorption peaks at 1,110 cm−1 were characteristic of the thermal degradation of lignocellulosic biomass, mainly generated by the vibrations caused by the expansion of C-O bonds (Wu et al., 2018). There was almost no change noted in the characteristic peaks of Fe modified/unmodified biochar after the adsorption of heavy metals (Fig. 5A), indicating that there was limited damage made to the lignocellulosic structure.

Figure 5 FTIR spectra of BC and Fe-BC500 (A), and MF (B) before and after adsorption of Cu and Cd.

The surface functional groups of BC detected at 1,700 cm−1, 1,560 cm−1, and 1,380 cm−1 were assigned to the carbonyl (C=O) (Wang et al., 2014). The C=C bending vibration, CH3 bending vibrations (Hao et al., 2021), and the hydroxyl (-OH) appeared at 1,420 cm−1 and 617 cm−1 (Bueno & Morgado, 1993; Mitić et al., 2009). The results demonstrated that the surface of the biochar from medicine residue included oxygen-containing functional groups, such as carboxyl, lactone, carbonyl, and hydroxyl. The strength of the framework-stretching vibration absorption peak of C=O in the aromatic ring at 1,700 cm−1 decreased after the adsorption of Cu and Cd. The characteristic peak for the -OH functional group of BC decreased at 1,420 cm−1, indicating that the hydroxyl groups possibly on the surface of the biochar may participate in the adsorption reaction process of Cu and Cd. Consequently, the BC surface, a type of porous material with the oxygen-containing functional groups mentioned above, was involved in the adsorption reaction process of Cu and Cd.

The absorption peak of CH3 bending vibrations at 1,380 cm−1 for Fe-BC was stronger than BC when comparing BC with Fe-BC before adsorption. The C =O absorption peak (1,700 cm−1), and -OH bending vibration peak (1,420 cm−1 and 617 cm−1) for Fe-BC practically disappeared. The absorption peak of Fe-OH at 796 cm−1 appeared (Chen et al., 2020), which is a characteristic peak for iron oxide, signaling that Fe was attaching on the surface of the biochar. The absorption peak at 796 cm−1 and 1,380 cm−1 significantly weakened after the adsorption of heavy metals with Fe-BC, implying that Fe-OH and -CH3 participated in the reaction process.

In addition, the FTIR spectra of Fe-BC samples (Fig. 5A) and maifanite samples after adsorption of Cu and Cd (Fig. 5B) showed a vibration peak of –OH group related to water at 3,380 cm−1 or 3,450 cm−1. Moisture was unavoidable in the experiments.

MF is rich in quartz, feldspar, and smectite, as aforementioned. The stretching vibrational peak corresponding to C=O and can be detected at a wavenumber of 1,800 cm−1, and the bending vibration peak of H-O-H at 1,630 cm−1. The stretching vibration absorption peak of Si-OH was detected at 1,030 cm−1. The FTIR absorption peaks intensities weakened after Cu and Cd adsorption, indicating the main components participated in the adsorption of heavy metals (Zhang, Luo & Zhang, 2012).

X-ray diffraction analysis of amendments before and after adsorption

Carbon was a main component in the biochar produced from the medicine residue of radix Isatidis. Patterns of XRD in Fig. 6A reveal a wide and slow diffusion peak in the range of 22°–36° for C (002), which is typically caused by the lamination of single atomic carbon layers in microcrystals. Moreover, C (101) of cellulose characteristic crystal plane diffraction peaks with relatively wide but small intensity was observed in the range of 12°–15°. We also detected characteristic peaks from SiO2 at 26.48°, CaCO3 at 29.56° and 43°, and CaMg(CO3)2 at 30.8° when comparing the resulting patterns with the standard pdf cards. The results showed the presence of SiO2, CaCO3, and CaMg(CO3)2 on the biochar surface when the biochar was formed in the process of gasification evaporation. Our results are similar to those of Wang et al. (2017), which detected alkaline salts, such as KCl, CaCO3, and MgCl2 in the biochar extracted from water hyacinth and maize straw.

Figure 6 XRD patterns of BC and Fe-BC (A) and MF (B) amendments before and after adsorption of Cu and Cd.

There were no clear changes in diffraction peak positions after the adsorption of Cu and Cd in BC were observed. C, SiO2, CaCO3, and CaMg(CO3)2SiO2 did not participate in Cu and Cd reactions. In addition, Cu2+ and Cd2+ were adsorbed on the surface of biochar with organic groups, such as hydroxyl, carboxyl, lactone, and carbonyl. We detected characteristic peaks of C, SiO2, CaCO3, and CaMg (CO3)2 SiO2 as well as magnetite in the X-ray diffraction pattern of Fe-BC (Fig. 6A). This observation indicates that the iron oxide loaded to BC might be magnetite.

The XRD peaks of maifanite after the adsorption of Cu and Cd (Fig. 6B) reveal the presence of SiO2, CaCO3, NaAlSi3O8 (sodium feldspar), and Al2Si2O5(OH)4 (smectite). The absorption peaks of smectite at 10.4° and 12.3° disappeared after adsorption of Cu and Cd. The absorption peaks of sodium feldspar at 8.75°, 13.8°, and 27.3° also weakened or disappeared completely as a result of Cu and Cd adsorption. These results indicate that the silica hydroxyl, hydroxyl, and carboxyl groups on the surface of sodium feldspar and smectite in maifanite participate in the adsorption of Cu and Cd. Analyses of SEM, FTIR, and XRD in the immobilization process are consistent.

Extraction fraction of heavy metals from amendments

BC, Fe-BC, and MF are mesoporous nanomaterials, which physically adsorbed Cu and Cd and generated a chemical complexation with functional groups on their surfaces. This affected the extraction fraction percentage of Cu and Cd across amendments.

The modified BCR sequential extraction method was used to analyze the Cu and Cd fraction from the amendments after adsorption (Fig. 7). The weak acid soluble fractions of Cu in MF, BC, and Fe-BC were 91.10%, 41.06%, and 52.65%, respectively. The reducible fractions of Cu were 1.59%, 10.03%, and 9.65%, respectively. The oxidizable fractions of Cu were 0.08%, 15.00%, and 11.98%, respectively, and the residual fractions of Cu were 7.23%, 33.92%, and 25.72%, respectively, after the adsorption equilibrium experiment. The residual fraction Cu in balanced MF with Cu and Cd solution was significantly lower than that in balanced BC and balanced Fe - BC, which was inconsistent with the results in treated soil. The residual fraction Cu in soil treated with MF was the highest among all of the amend treatments (Fig. S1). The pH of the solution in the adsorption balance experiment was affected by the amendments. The pH of MF (7.03) was neutral and lower than that of BC (10.93) and Fe-BC (8.44) (Table 1). The immobile effect of MF on heavy metals was worse than that of BC and Fe-BC. The soil pH was 6.30, 5.99 and 5.77, respectively, after being treated with the test amendments. The change of pH in test soils was not consistent with the solution, in which MF increased the soil pH most significantly. Thus the increase in heavy metal residual fraction in the soil was the most significant among the three amendments.

Figure 7 The heavy metal extraction fraction percentage in amendments after adsorption of Cu (A) and Cd (B).

The weak acid soluble fractions of Cd in MF, BC, and Fe-BC were 97.24%, 86.14%, and 84.38%. The reducible fractions of Cd were 0.48%, 12.09%, and 13.18%. The oxidizable Cd fractions were undetected. The residual fractions of Cd were 2.28%, 1.16% and 1.82%, respectively, after the adsorption equilibrium between amendments and the Cu, Cd solution (Fig. 7B). The residual fractions of heavy metals in soil were generally not absorbed by the plants. The residual fraction of Cu in all treatments was far higher than Cd, indicating that the immobile effect of the three amendments on Cu was better than that on Cd. These results are consistent with the content of Cu and Cd in ryegrass after treatment.

Extraction fraction of heavy metals from soil colloid with amendments

The heavy metals and the soil colloid interacted with the amendment in the soil environment. Soil colloids are the basic soil functional unit carrying heavy metals due to its small particle size, larger SSA and negative charge. It was also a key factor affecting the distribution of heavy metals in the soil. Therefore, the total content of Cu and Cd in colloids was much higher than that in the soil (Table 2). For example, the content of Cu and Cd in untreated polluted soil colloids was 489.92 mg kg−1 and 2.57 mg kg−1, respectively, which was 2.04 and 1.33 times that of soil Cu (239.98 mg kg−1) and Cd (1.93 mg kg−1). The accumulation effect on Cu is greater than Cd in colloid.

Table 2 Effect of amendments on the content of Cu and Cd in soil colloid.

Treatment	CK	MF	BC	Fe-BC	
Cu/mg kg−1	489.9 ± 8.4a	488.1 ± 11.4a	500.5 ± 24.2a	471.0 ± 8.8a	
Cd/mg kg−1	2.57 ± 0.04b	2.17 ± 0.23a	2.35 ± 0.53a	1.98 ± 0.06a	

Heavy metals in soil colloids can be divided into an amorphous iron oxide bound fraction, a crystal iron oxide bound fraction and a residual bound fraction. It is well known that the increase of the amorphous iron oxide bound fraction signals the decrease in the phytoavailability of heavy metals. The forms of Cu and Cd in the soil colloid after treated with MF, BC and Fe-BC were all relocated (Fig. 8). Compared with the CK treatment, the content of amorphous/non-crystalline iron oxide-bond Cu in colloids after immobilization treatments increased significantly by 20.34%, 16.81%, and 18.87%, respectively. The content of residual bond Cu decreased significantly by 20.23%, 17.01%, and 19.17%, respectively. Non-crystalline iron oxide bond Cd content increased significantly by 3.72%, 4.18%, and 6.58%, respectively, and the residual bond Cd decreased significantly by 3.65%, 4.57%, and 7.04%, respectively. The variation in the crystalline iron oxide bond of Cu or Cd was not significant. The effect of amendments on Cu was greater than Cd. Moreover, soil endogenetic minerals and organic matter may participate in the immobilization process, affecting the redistribution of heavy metals in the soil and soil colloid.

Figure 8 Effects of different amendments on the extract fraction of Cu (A) and Cd (B) in soil colloid.

Conclusions

The studied amendments, BC, Fe-BC, MF, BC:MF(1:1), and Fe-BC:MF(1:1), effectively immobilized copper and cadmium, and the types and combinations of BC and MF used affected Cu and Cd differently. Amendments were better able to immobilize Cu than Cd. The Cu content in shoots significantly decreased from 209.3 mg kg−1 (CK) to 51.50 mg kg−1 (BC), 57.90 mg kg−1 (Fe-BC:MF), 59.29 mg kg−1 (BC:MF), 63.74mg kg−1 (MF), and 70.92 mg kg−1 (Fe-BC). Meanwhile, the Cd content decreased from 2.06 mg kg−1 (CK) to 1.43 mg kg−1 (MF), 1.46 mg kg−1 (BC), 1.51 mg kg−1 (BC:MF), 1.53 mg kg−1 (Fe-BC), and 2.01 mg kg−1 (Fe-BC:MF).

BC, Fe-BC, and MF may physically adsorb copper and cadmium. They may also generate chemical complexation with functional groups which could then affect the extraction fraction percentage of Cu and Cd in the amendments. The content of amorphous crystalline iron oxide bound Cu in colloid was higher than Cd, and was a key step to immobilization.

The studied amendments can be used to enhance the immobilization of copper and cadmium in the contaminated paddy soils. The immobilization effects of the amendments depended on the metal element and combination of amendments. The suitable amendment method outlined in this research may be used to prevent the contamination of heavy metals in soils, especially Cu and Cd.

Supplemental Information

Supplemental Information 1 The content of Cu, Cd in underground parts of plants and the original data of BCF and TF

The file is the raw data in Figs. 1–3. The file contains the content of heavy metals in the underground part root of plants in Fig. 1, the BCF in Figure 2 and the TF in Fig. 3.

Click here for additional data file.

Supplemental Information 2 Basic properties of biochar and maifanite

This data is the original data of Table 1.

Click here for additional data file.

Supplemental Information 3 Content of Cu and Cd in soil colloid

This data is the original data of Table 2.

Click here for additional data file.

Supplemental Information 4 Original data of heavy metal speciation in amendments

The file is the raw data of Fig. 7.

Click here for additional data file.

Supplemental Information 5 Original data of heavy metal speciation in soil colloids

The file is the raw data of Fig. 8.

Click here for additional data file.

Supplemental Information 6 Figure S1. Changes of heavy metal extraction fraction percentage in soil treated with immobilizers

Changes of heavy metal extraction fraction percentage in soil treated with immobilizers.

Click here for additional data file.

Additional Information and Declarations

Competing Interests

Author Contributions

Data Availability

The authors declare there are no competing interests.

Yuan Ding conceived and designed the experiments, authored or reviewed drafts of the paper, and approved the final draft.

Weiya Wang and Shiying Ao performed the experiments, analyzed the data, prepared figures and/or tables, and approved the final draft.

The following information was supplied regarding data availability:

The raw data are available in the Supplemental Files.

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
