# Peer review of "The regulating effects and mechanism of biochar and maifanite on copper and cadmium in a polluted soil-Lolium perenne L. system"

_PeerJ, doi:10.7717/peerj.11921_

## Round 0.1 · original submission · Minor Revisions

Please consider the suggestions from the reviewers carefully, especially regarding the language and FTIR interpretation. The language could be improved by the service of a professional editor. Also, please consider revisiting the title that is currently long and difficult to understand (.e.g. is not clear what "their" refers to).

Reviewer 1 ·

Basic reporting

no comment

Experimental design

no comment

Validity of the findings

no comment

Additional comments

This paper includes the valuable results obtained experimental studies to remove heavy metals from the polluted soil using different amendments. In order to remove copper and cadmium from contaminated soil, biochar, iron modified biochar, maifanite have applied, and the mechanisms of copper and cadmium removal of soil particles have investigated. The effect of the application on the prevention of Cu and Cd accumulation in the shoots and roots of ryegrass has investigated. According to the results obtained experimental studies, the immobile effects of examined amendments on Cu has found better than Cd. In addition to the adsorption mechanism, complex formation occurred on biochar, iron modified biochar, maifanite. The results of experimental data have a great importance to the remediation of heavy metal polluted soils.
The paper is properly well organised and overall quality of the work is excellent. The paper is well within the scope of the journal it needs some minor improvements and revisions given below:
Avoid using words in first person, like: we, I.
In the Conclusions part at lines 374- 377 the sentence needs to be improved.

Reviewer 2 ·

Basic reporting

The theme of the manuscript is interesting and well organized.
The author has often used conjunctions in starting a sentence which needs to be corrected and should also focus on grammatical errors.

For example:
1. lines 227-229: And for plants which enter the food chain, this indicator is much more
important because it fully expresses the total amount of heavy metals uptaken from
the soil.
2. Line 295-298: Few change was detected in the characteristic peaks of Fe
modified/unmodified biochar after the adsorption of heavy metals (Fig. 5A), indicating
limited damage on the structure of cellulose and hemicellulos

For more such comments, refer to the attached review document.


Some of the references cited in the paper are not adding any value or supporting the context.

For example:
1. In lines 299-303 it was stated that "As for BC, the surface functional groups were
apparently detected at 1700 cm-1, 1560 cm-1, 1420 cm-1, 1380 cm-1, and 617 cm-1;
assigned to carbonyl (C=O) and the hydroxyl (-OH) groups, demonstrating that the
surface of biochar from medicine residue included oxygen-containing functional
groups, such as carboxyl, lactone, carbonyl, and hydroxyl (Foo & Hameed, 2010)."

but the cited reference is neither discussing about FT-IR nor about the functional group characterization.

For more such comments, refer to the attached review document.

Experimental design

no comment

Validity of the findings

In lines 305-308, author has stated that "The characteristic peak for –OH functional group of BC decreased at 1420 cm-1, indicating that the hydroxyl groups possibly contained on the surface of the biochar may participate in the adsorption reaction process of Cu and Cd."

but in reality a broad peak around 3500-3000 cm-1 confirms the presence of hydroxy functional group (-OH). This is visible in Fe-BC (3380 cm-1) whereas missing in BC.
Can the author explain the reason behind this?

In line 313-315, author has stated that "And absorption peak of Fe-OH at 796 cm-1 appeared, which is a characteristic peak for iron oxide, signaling Fe attaching on the surface of the biochar."

Author has to provide academic reference to support that a peak at 796 cm-1 characterizes the presence of Iron oxide.

Additional comments

The current paper is well organized. Through the abstract, the author clearly summarized major aspects of the paper such as the key purpose of the study, research problems, basic design of the study, and key findings. The introduction and background are reasonable given the premise of the manuscript submitted. All the tables and figures provided are comprehensive and clearly represent the experimental findings.

Please respond to the comments and questions regarding the FT-IR analysis that was conducted to find the variations in functional groups of amendments before and after adsorption of Cu and Cd.

Annotated reviews are not available for download in order to protect the identity of reviewers who chose to remain anonymous.

---

## Round 0.2 · accepted · Accept

Thank you for addressing the reviewers' concerns adequately.